# Implementation of BFASTmonitor Algorithm on Google Earth Engine to Support Large-Area and Sub-Annual Change Monitoring Using Earth Observation Data

**Eliakim Hamunyela [1], Sabina Rosca [2], Andrei Mirt [2] , Eric Engle [3], Martin Herold [2], Fabian Gieseke [4] and Jan Verbesselt [2,*]**

[1] Faculty of Humanity and Social Sciences, University of Namibia, Private Bag 13301, Windhoek 10000, Namibia; hamunyelae@unam.na

[2] Laboratory of Geo-Information Science and Remote Sensing, Wageningen University & Research, Droevendaalsesteg 3, 6708 PB Wageningen, The Netherlands; sabina.rosca@wur.nl (S.R.); andrei.mirt@wur.nl (A.M.); martin.herold@wur.nl (M.H.)

[3] Google Inc., 1600 Amphitheater Parkway, Mountain View, CA 94043, USA; eengle@google.com

[4] Department of Information Systems, University of Münster, 48149 Münster, Germany; fabian.gieseke@uni-muenster.de

**\*** Correspondence: jan.verbesselt@wur.nl; Tel.: +31-317-485-268

**Abstract:** Monitoring of abnormal changes on the earth's surface (e.g., forest disturbance) has improved greatly in recent years because of satellite remote sensing. However, high computational costs inherently associated with processing and analysis of satellite data often inhibit large-area and sub-annual monitoring. Normal seasonal variations also complicate the detection of abnormal changes at sub-annual scale in the time series of satellite data. Recently, however, computationally powerful platforms, such as the Google Earth Engine (GEE), have been launched to support large-area analysis of satellite data. Change detection methods with the capability to detect abnormal changes in time series data while accounting for normal seasonal variations have also been developed but are computationally intensive. Here, we report an implementation of BFASTmonitor (Breaks For Additive Season and Trend monitor) on GEE to support large-area and sub-annual change monitoring using satellite data available in GEE. BFASTmonitor is a data-driven unsupervised change monitoring approach that detects abnormal changes in time series data, with near real-time monitoring capabilities. Although BFASTmonitor has been widely used in forest cover loss monitoring, it is a generic change monitoring approach that can be used to monitor changes in a various time series data. Using Landsat time series for normalised difference moisture index (NDMI), we evaluated the performance of our GEE BFASTmonitor implementation (GEE BFASTmonitor) by detecting forest disturbance at three forest areas (humid tropical forest, dry tropical forest, and miombo woodland) while comparing it to the original R-based BFASTmonitor implementation (original BFASTmonitor). A map-to-map comparison showed that the spatial and temporal agreements on forest disturbance between the original and our GEE BFASTmonitor implementations were high. At each site, the spatial agreement was more than 97%, whereas the temporal agreement was over 94%. The high spatial and temporal agreement show that we have properly translated and implemented the BFASTmonitor algorithm on GEE. Naturally, due to different numerical solvers being used for regression model fitting in R and GEE, small differences could be observed in the outputs. These differences were most noticeable at the dry tropical forest and miombo woodland sites, where the forest exhibits strong seasonality. To make GEE BFASTmonitor accessible to non-technical users, we developed a web application with simplified user interface. We also created a JavaScript-based GEE BFASTmonitor package that can be imported as a module. Overall, our GEE BFASTmonitor implementation fills an important gap in large-area environmental change monitoring using earth observation data.

**Keywords:** Google Earth Engine; BFASTmonitor; forest disturbance; sub-annual; Landsat

## 1. Introduction

The earth continues to experience many adverse changes as a result of climatic and anthropogenic pressures. For example, the tropical forests, which are the "lungs" of the earth system and play a critical role in regulation of the global climate [1,2], are increasingly under human-induced pressure through deforestation and forest degradation [3,4]. Tropical forests are being deforested mainly to expand agricultural land [5], and are degraded through selective logging [6]. In some cases, deforestation and forest degradation activities are carried out illegally, often to fulfil commercial interests. Satellite remote sensing has been instrumental in revealing how tropical forests are being deforested and degraded across the globe [3,4,7], and is now used for frequent and large-area monitoring of forest changes to provide timely information on forest disturbances [8,9]. Several operational systems for sub-annual forest monitoring have been developed (e.g., Global Forest Watch: https://www.globalforestwatch.org/), and have been helpful in detecting forest disturbances in a timely manner in humid tropical forests [8,9], but the monitoring systems are often not robust in regions where forests exhibit strong seasonal variability in their photosynthetic activity and canopy water content [10]. This is mainly because the monitoring systems rely on change detection methods, which are unable to account for sub-annual seasonal variations in forest condition. Accounting for sub-annual seasonal variations is especially important as the seasons are changing more rapidly from year to year, and forests are experiencing more extreme weather events, such as more frequent and intense droughts [11–13]. Such extreme events are likely to trigger frequent and intense changes in the earth's system at various spatio-temporal scales, and understanding such changes will require development of tools that provide efficient sub-annual monitoring capabilities over large area.

Change detection methods, for example BFASTmonitor (Breaks For Additive Season and Trend monitor [14]), CCDC (Continuous Change Detection and Classification [15]) and STEF (Space-Time Extremes and Features [16]), which are able to account for sub-annual seasonal variation while detecting forest disturbances from satellite observations have been developed in recent years to support automated sub-annual forest monitoring in both humid and dry forest ecosystems. Despite the benefits these methods bring to forest monitoring, they are not yet integrated into operational forest monitoring systems; they remain confined to the research domain or only available in highly specialised computational platforms which are too challenging for many current and potential users.

High computational costs often inhibit forest monitoring at sub-annual scale over large areas. Recently, however, computationally powerful platforms, such as the Google Earth Engine (GEE), have been launched to support planetary analyses of satellite data for environmental monitoring [17]. With GEE, computationally demanding global analyses of forest loss and gain [4], tidal flats [18], and surface water dynamics [19] at unprecedented spatial scale (30m spatial resolution) became possible for the first time. This increase in computing power, together with state-of-the-art algorithms, can also be exploited to monitor forest changes at sub-annual scales using dense time series from satellite with improved spatial resolution (e.g., Landsat, Sentinel-1 and-2) to improve the detection of deforestation and forest degradation, especially in the tropics.

Some time series analysis methods for forest disturbance detection have been implemented in the GEE recently (e.g., [20]), but the methods are only developed for the detection of forest disturbances at annual scale, and not at sub-annual scale [21,22]. The implementation of methods for sub-annual scale forest monitoring has been slow, probably because of the complexity of the methods requiring a combination of iterative robust statistics and capacity to be able to analyse and model seasonal variation in the time series. To fill this gap, we implemented the BFASTmonitor algorithm [14] to support large-area and sub-annual forest monitoring in both humid and dry tropical forests.

BFASTmonitor is a data-driven unsupervised change monitoring approach that detects abnormal changes in time series data (See Figure 1). Abnormal changes in the time series are detected based on a structural change monitoring framework [23–25]. BFASTmonitor was developed specifically for near real-time change detection of abnormal changes in time series data [14]. When new measurements are available, the algorithm verifies whether the new observations are normal or abnormal based on the modelling of the already available time series (i.e., the history). The regression coefficients are first estimated from the historical observations and are subsequently used to predict the values of observations in the monitoring period [14,23–25]. An abnormal change is signalled if the predicted values are statistically different from observed values [14]. The versatility of the BFASTmonitor to detect abnormal changes at sub-annual scale has been demonstrated in various forest ecosystems [10,26,27] to detect forest cover loss, but the demonstration has largely been restricted to small areas (smaller than one Landsat scene of 32,400 km$^2$) due to the high computational costs. Yet, forest disturbances and other changes on earth surface occur simultaneously across the globe. Therefore, to better understand such changes and their impact, we require tools and systems with capabilities for large area monitoring and analysis. Capacity for large area monitoring can, for example, benefit countries that want to protect their forests from deforestation and human-induced forest degradation.

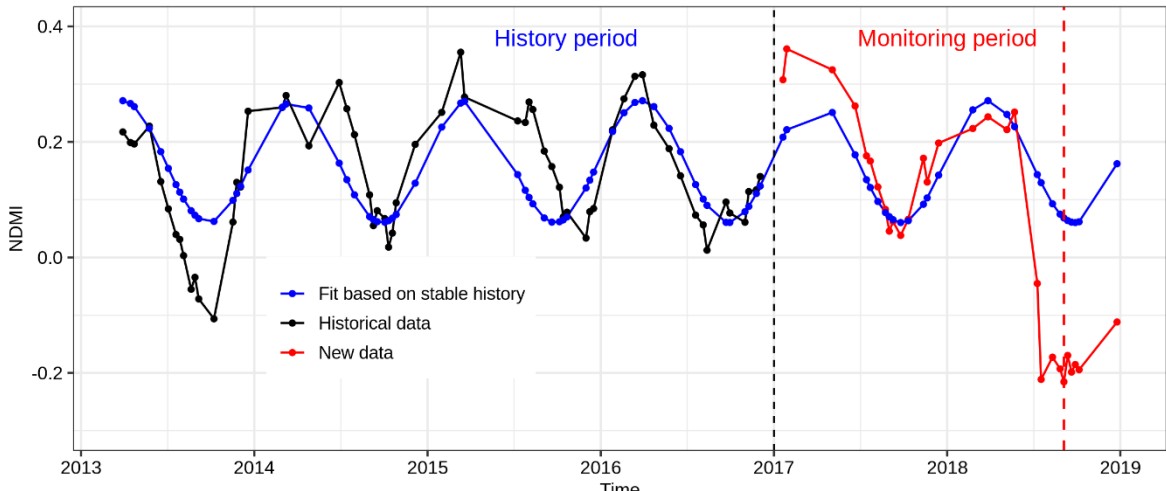

**Figure 1.** An example of how Breaks For Additive Season and Trend monitor (BFASTmonitor) uses historical data in the time series to identify abnormal changes in the monitoring period. Regression models estimated from the historical observations are used to predict the values of the new observations in the monitoring period. An abnormal change is signalled if the new observations are statistically different from the predicted values and the magnitude of change is negative. The vertical black dashed line represents the start of the monitoring period, whereas the vertical red dashed line indicates an abnormal change detected by BFASTmonitor. (BFAST package [14]). Here the observations are for normalised difference moisture index (NDMI) derived from Tier 1 Landsat-7/ETM+ and Landsat-8/OLI surface reflectance products.

In this paper, we report on the implementation on BFASTmonitor on GEE to support sub-annual monitoring of abnormal changes on the earth surface over large areas using satellite observations. We demonstrate the performance of GEE BFASTmonitor by applying it to three study areas while comparing it with the original R-based BFASTmonitor implementation [14]. The performance of the GEE BFASTmonitor was evaluated using the time series of Landsat normalised difference moisture (water) index (NDMI; [28]). A web application with a simplified user-interface has been developed to make GEE BFASTmonitor accessible to, and easy-to-use for, non-technical users.

Prior to our implementation, the BFASTmonitor algorithm was only available as part of the BFAST R package and as an implementation in python, deployable on the graphics processing unit (GPU) [29]. This implementation supports processing of large amounts of satellite data on local desktop computers that have GPU's available. However, this implementation still requires the data to be downloaded to a local computer. With GEE implementation, we are bringing BFASTmonitor algorithm closer to large amounts of data, thus avoiding the limitations associated with downloading massive amount of data to a local computer.

## 2. Materials and Methods

### 2.1. BFASTmonitor Implementation on Google Earth Engine

We implemented the BFASTmonitor approach on GEE by translating the original BFASTmonitor code (http://bfast.r-forge.r-project.org/ [14,30]) to a corresponding JavaScript version using the GEE editor. Fortunately, the GEE libraries already provide many basic functions and regression model fitting routines, which are critical for the implementation of the BFASTmonitor. In our implementation, the users have a choice to fit a regression model with or without seasonal terms based of ordinary least squares (OLS). The trend component can also be included. The regression model with seasonal terms takes the following form:

$$\hat{a}_t \; = \; \beta_0 + \sum_{j=1}^{k} \gamma_j \sin\left(\frac{2\pi j t}{f} \; + \; \delta_j\right) + \; \varepsilon_t \tag{1}$$

Here, $\hat{a}_t$ is the predicted value of $\hat{a}_t$ at time t, $\beta_0$ is the intercept coefficient estimated from observations in the history period, $\gamma_j$ is the amplitude, $\delta_j$ is the phase of the harmonic season and $f$ is the frequency, corresponding to the number of observations expected per year at each pixel, $\varepsilon_t$ is the random error at time $t$ [14,30,31]. With the BFASTmonitor, the number of the harmonic terms (k) has to be specified. In our implementation, the users have the choice of specifying the number of harmonic terms to fit.

The functionality for testing and detecting structural changes in the time series was lacking in GEE. We implemented this functionality in the GEE afresh. The original BFASTmonitor offers several approaches for testing and detecting structural breaks in the time series [14]. The default and widely used approach is the OLS based moving sum (OLS-MOSUM) of residuals, which we considered for our GEE implementation. The OLS-MOSUM of residuals is calculated as follows:

$$MO_t \; = \; \frac{1}{\sqrt[\sigma]{n}} \sum_{s=t-h+1}^{t} \left(y_s \; - \; \tilde{y}_s\right) \tag{2}$$

Here, $\sigma$ is an estimator of the variance, $n$ is the number of observations in the history period, $h$ is the size of the moving window, which is calculated as a fraction of $n$, and $t$ is time [23,25]. Without a structural break in the time series, the MOSUM of residuals ($MO_t$) is expected to remain within the boundary of its empirical fluctuation [23]. The empirical fluctuation of MOSUM residuals is the historical oscillation of MOSUM of residuals, based on the observations in the history period. A threshold (alpha) is needed for accepting or rejecting the null hypothesis that the MOSUM of residuals did not cross the boundary of its empirical fluctuation. Alpha is the significance level (e.g., 5%), and users have a choice to specify it. A breakpoint is signalled once the boundary is crossed by MOSUM of residuals. Once a breakpoint is detected, we calculate the magnitude of change (mc) by subtracting the predicted value $y_t$ from observation $y_t$ at time $t$ when a breakpoint is detected.

BFASTmonitor detects both the positive and negative abnormal change in the time series. Our GEE implementation also detects both positive and negative abnormal changes. A positive break in the NDMI time series over a forest area may signal an increase in the forest cover or productivity, whereas the negative break may indicate forest disturbance. However, outliers related to atmospheric

contamination (e.g., remnants clouds) missed by cloud masking algorithm might also be detected as breakpoints in the time series of the optical satellite data. Detailed information about BFASTmonitor are available in [14], whereas more detailed information about structural break detection approaches can be found in [23,25].

### 2.2. Making GEE BFASTmonitor Accessible

Implementing a state-of-the-art algorithm like BFASTmonitor on GEE is a great step towards an improved the near real-time monitoring of changes on the earth's surface over large areas using time series of satellite observations. However, such state-of-the-art monitoring capabilities may not be accessible to non-technical users who may be interested in exploring and documenting earth's surface changes occurring in their areas. To fill this gap, in addition to the GEE BFASTmonitor package, we developed a web application with simplified user interface to make the GEE BFASTmonitor accessible for non-technical users. We also created a GEE BFASTmonitor package that can be imported into GEE to make it easier for advanced GEE users to integrate our algorithm in their processing chain. The web application and the package are written in GEE's JavaScript API (application programming interface).

### 2.3. Evaluating GEE BFASTmonitor

We evaluated our GEE BFASTmonitor implementation by comparing its forest disturbance detection performance to that of the original BFASTmonitor [14]. Here, we defined forest disturbance as an abnormal change in the time series with a negative magnitude of change over a forested area. The evaluation was conducted at three forest sites (Table 1), located in Peru, Bolivia and Mozambique (Figure 2). The site in Peru is a humid tropical forest located in Madre de Dios province, south-eastern Peru. This site is part of the Tropical Andes biodiversity hotspot [32], but has been experiencing rapid forest change because of the cropland expansion and gold mining [33,34]. The forest at the Peruvian site does not have defined seasonal variability in its photosynthetic activity and canopy water content (Figure 1). In contrast, the site in Bolivia is a dry tropical forest, with a moderate seasonality in its photosynthetic activity and canopy water content (Figure 3b). The Bolivian site is located in the south east of Santa Cruz. Forest disturbances at this Bolivian site are generally large, caused by clearance of forest to establish large-scale agriculture [10]. Unlike the other two sites, the Mozambican site is a Miombo woodland, with strong seasonal variations in the photosynthetic activity and canopy water content (Figure 3c). At this site, forest disturbance is dominated by small-scale disturbances caused mainly by smallholder farmers. The Mozambican site is located in one of the frequently burned areas in southern Africa. We chose these three study sites because they cover a variety of tropical forest types. They also have varying forest disturbance types caused by different agents. These sites are therefore appropriate for an elaborate assessment of the GEE BFASTmonitor performance under varying conditions.

**Table 1.** The size, forest type, and the number of Landsat images available for each study site.

| Site | Forest Type | Size (km$^2$) | Number of ImAges in Reference Period | Number of Images in Monitoring Period | Total Number of Images |
|------|-------------|---------------|--------------------------------------|---------------------------------------|------------------------|
| Bolivian | Dry tropical forest | 10,112 | 113 | 60 | 173 |
| Peruvian | Humid tropical forest | 5274 | 102 | 47 | 149 |
| Mozambican | Miombo woodland | 15,569 | 118 | 60 | 181 |

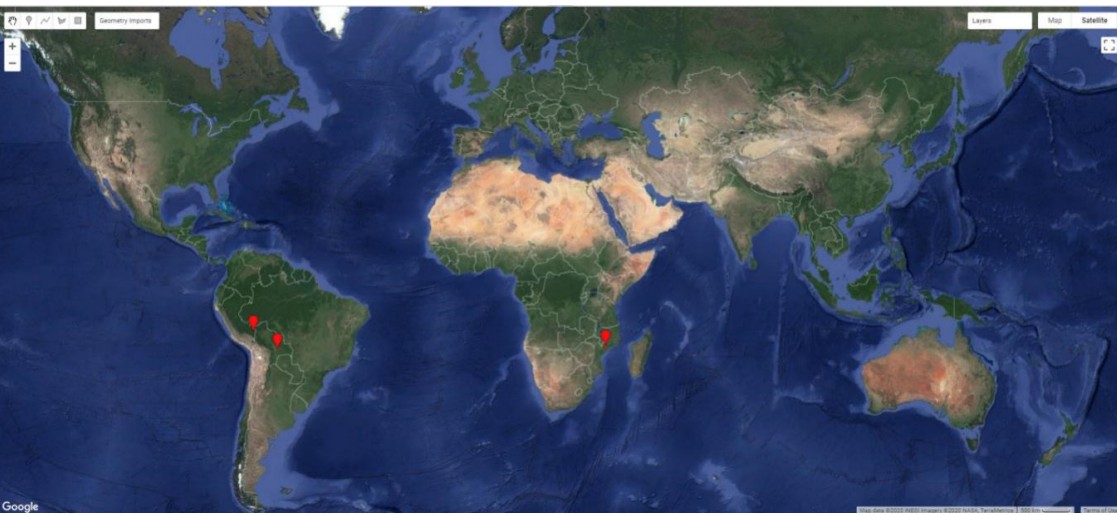

**Figure 2.** Location of the study sites. The background image is from Google Earth™.

At each site we detected forest disturbance from the Landsat NDMI time series using original and GEE BFASTmonitor. We detected forest disturbances between January 2017 and December 2018 (the monitoring period) while using the 2014–2016 Landsat NDMI time series as a reference (the history period). The NDMI time series were derived from Tier 1 Landsat-7/ETM+ and Landsat-8/OLI surface reflectance products available on GEE. Tier 1 Landsat data are of high quality and are considered suitable for time series analyses. Landsat-7/ETM + surface reflectance products on GEE were generated using the standard Landsat Ecosystem Disturbance Adaptive Processing System (LEDAPS) algorithm [35,36], whereas the Landsat-8/OLI surface reflectance products were generated using the Landsat-8/OLI surface reflectance algorithm [37]. We masked clouds, cloud shadows and water bodies using pixel quality flags that accompany Landsat surface reflectance products. The default pixel quality flags of Landsat data were generated using the Cmask algorithm [38,39]).

To compare with the original BFASTmonitor consistently, we downloaded and pre-processed the same Landsat NDMI time series from GEE. The comparison test was run on a Linux machine with 16 GB of RAM and 8 CPU cores. We used the same values for the harmonic term, h-parameter and the alpha when running original and GEE BFASTmonitor at each site. By default, the original BFASTmonitor calculates the magnitude of change by subtracting the predicted value at the breakpoint from median value of all observed values after the breakpoint. This way of calculating the magnitude of change could lead to some forest disturbances having a positive magnitude of change. Forest disturbances with positive magnitude of change would be difficult to distinguish from breakpoints related to the increase in forest cover or productivity. To ensure comparability, we calculated the magnitude of change from the original BFASTmonitor implementation using the approach used for GEE BFASTmonitor. The harmonic terms and h-parameter are the most critical parameters that should be set when applying BFASTmonitor. For each site, we used one harmonic term (harmonic = 1), h-parameter of 0.05 (h = 0.25) and the alpha of 0.05 when applying GEE BFASTmonitor. The h-parameter ranges between 0 and 1, and determines the size of the moving window for MOSUM relative to the number observations in the history period. The original R-based BFASTmonitor implementation uses the default value of 0.25 for h-parameter. A h-parameter of 0.25 implies that the size of the moving window is equal 25% of observations in the history period. We used all NDMI observations in the history period for estimation of regression coefficients. The output was the time of disturbance and the magnitude of change. The maps of the time and magnitude of change were then exported from the GEE to a local computer, for further processing.

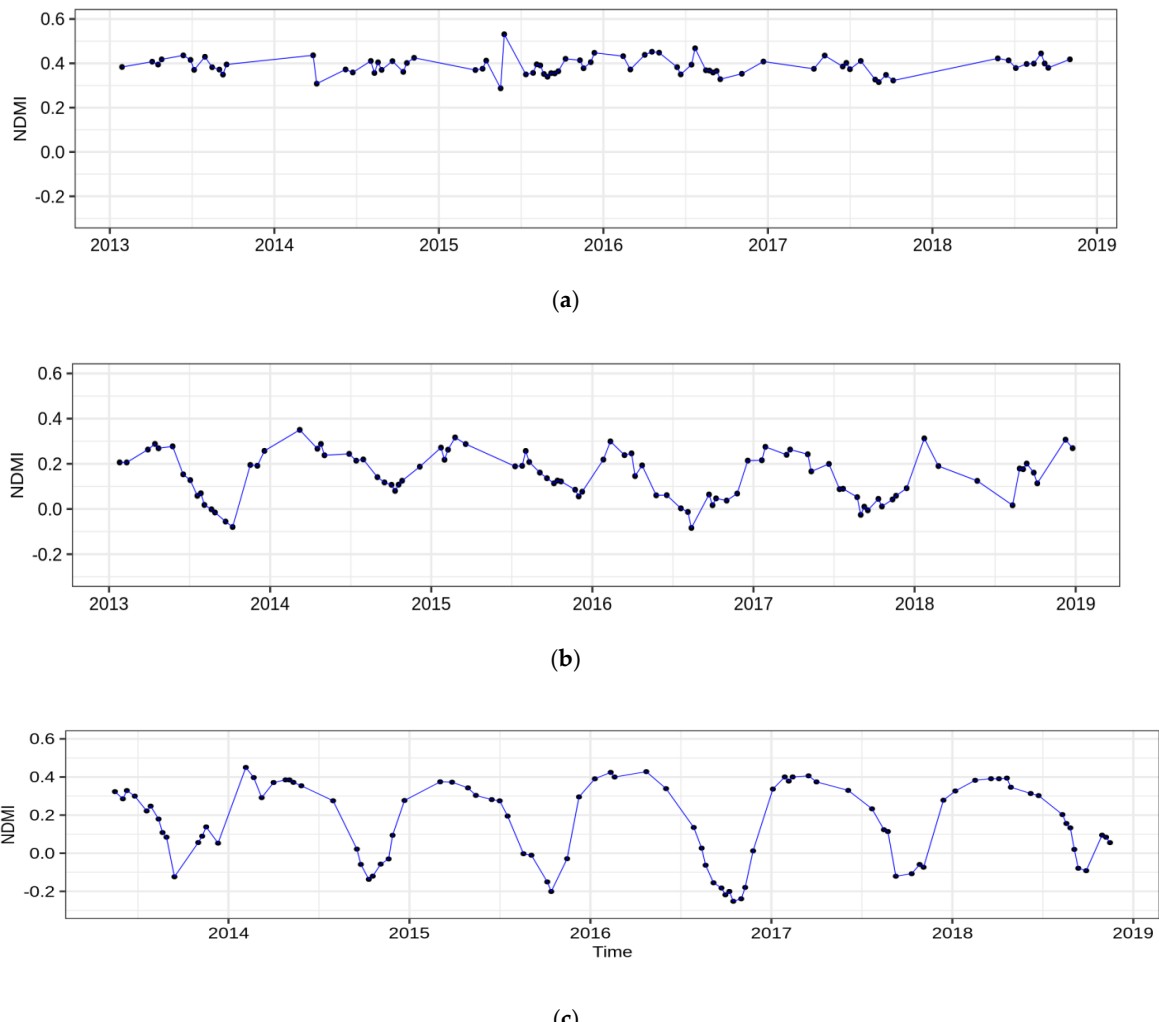

**Figure 3.** An example of Landsat pixel-time series for normalised difference moisture index (NDMI) of a (**a**) humid (Peruvian site), (**b**) dry (Bolivian site) tropical forest site and (**c**) miombo woodland (Mozambican site). The time series are for undisturbed pixels.

We performed a map-to-map comparison to determine the spatial agreement between the outputs from the BFASTmonitor and GEE BFASTmonitor. The primary goal for implementing BFASTmonitor on GEE is to support large-area monitoring of forest disturbances. Therefore, our comparison of original and GEE BFASTmonitor focused on structural breaks in NDMI which had negative magnitude of change. Negative magnitude of change implies that the predicted value at a structural break is greater than the observed value. Therefore, prior to comparison, we masked all pixels with positive magnitude of change in the outputs of both implementations. We also applied a benchmark forest mask to both outputs to mask changes that occurred in non-forest areas. A benchmark forest mask was created for each study site through supervised classification of the 2016 Landsat images using random forests [40]. The generation of the benchmark forest masks and the masking non-forest areas were performed on a local computer. We ensured that, for each site, the outputs from BFASTmonitor and GEE BFASTmonitor were covering exactly the same spatial extent. Henceforth, we refer to forest changes with negative magnitude of change as forest disturbance.

A discrete map was created from each implementation to show disturbed and undisturbed forest areas at each study site. The discrete maps at each study site were then combined into one map to show areas of spatial agreement and disagreement between original and GEE BFASTmonitor implementations. Overall, the spatial agreement between the implementations was calculated by dividing the total of number of pixels where both implementations agree spatially by the total number

of disturbed pixels at each study area based on a combined discrete map. The temporal agreement was calculated by dividing the total number of pixels that have a temporal gap of zero by the total number of pixels, where the original and GEE BFASTmonitor agreed spatially. The temporal gap was calculated by subtracting the date of forest disturbance as per GEE BFASTmonitor from that of original BFASTmonitor. We also compared the magnitudes of change from the original and GEE BFASTmonitor for temporally agreeing pixels by calculating the Pearson correlation coefficient between them.

## 3. Results

### 3.1. A Web Application with Simplified User Interface

Our easy-to-use GEE web application (TerraSift) with a simplified user-interface (Figure 4) make GEE BFASTmonitor accessible to non-technical users. TerraSift change detection is powered by the GEE BFASTmonitor and is currently accessible via: https://andreim.users.earthengine.app/view/bfastmonitor. The users can adjust BFASTmonitor parameters (i.e., start and end dates for the history and monitoring period, the h-parameter, number of harmonic terms to fit, the alpha and the period). Additionally, users can inspect the time series for a specific pixel by clicking on the map. TerraSift generates two raster layers (date and magnitude of change) and these layers can also be downloaded for further processing or restyling. With TerraSift, the state-of-the-art satellite imagery-based change detection is available for a broader public. Advanced users who may wish to integrate BFASTmonitor algorithm in their GEE processing chains, can use the GEE BFASTmonitor package (https://github.com/bfast2/geeMonitor).

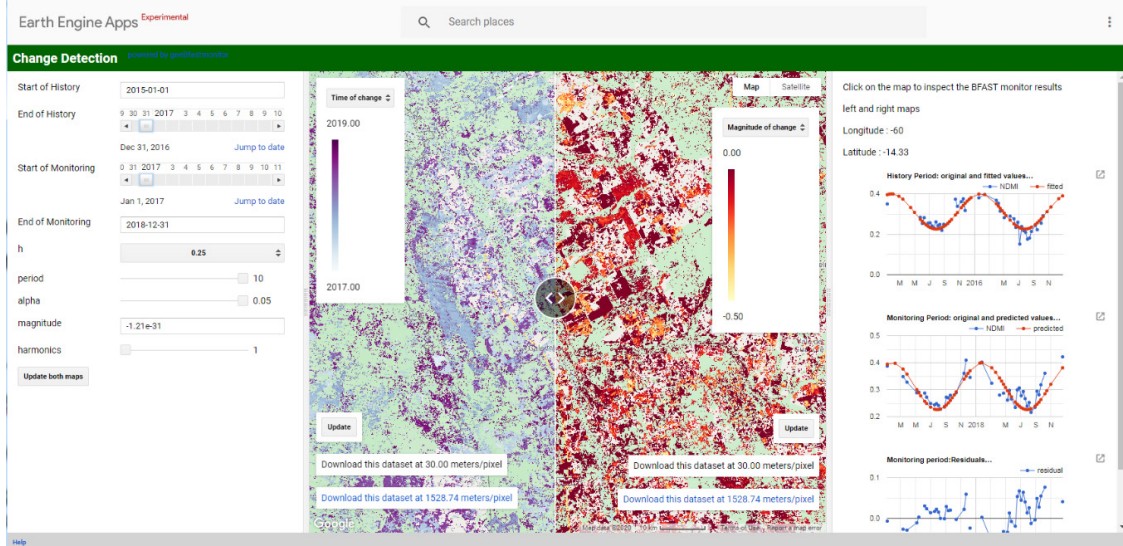

**Figure 4.** A screenshot of the user-interface for the web application (TerraSift) with forest disturbances detected by Google Earth Engine (GEE) BFASTmonitor from the time series of Landsat normalised difference moisture index at a site in Bolivia.

### 3.2. Comparison of Original and GEE BFASTmonitor on Forest Disturbance Detection

The spatial and temporal agreement between the original and GEE BFASTmonitor on forest disturbance detection were generally high at each study site (Table 2). At each site, the spatial agreement was more than 97%, whereas the temporal agreement was over 94%. High spatial and temporal agreement indicates that the BFASTmonitor algorithm was implemented properly in GEE. There were, however, some noticeable spatial and temporal disagreements between original and GEE BFASTmonitor (Figure 5), despite using the same parameters (harmonic terms, h-parameter and alpha) and calculating the magnitude of change in the same manner. The differences were more noticeable at the sites (Mozambican and Bolivian site), where forest has strong seasonal variability

in its photosynthetic activity and canopy water content. There were pixels, at each site, where the original BFASTmonitor detected disturbance earlier than GEE BFASTmonitor, and vice versa. However, the temporal gap was not more than 2 months (60 days) in more than 85% of such pixels (Figure A1).

**Table 2.** Spatial and temporal agreement between original and GEE BFASTmonitor on forest disturbance detected from Landsat NDMI time series at each site.

| Site | Spatial Agreement (%) | Temporal Agreement (%) |
|------|----------------------|------------------------|
| Bolivian | 97.8 | 94.8 |
| Peruvian | 99.6 | 99.5 |
| Mozambican | 97.8 | 97.7 |

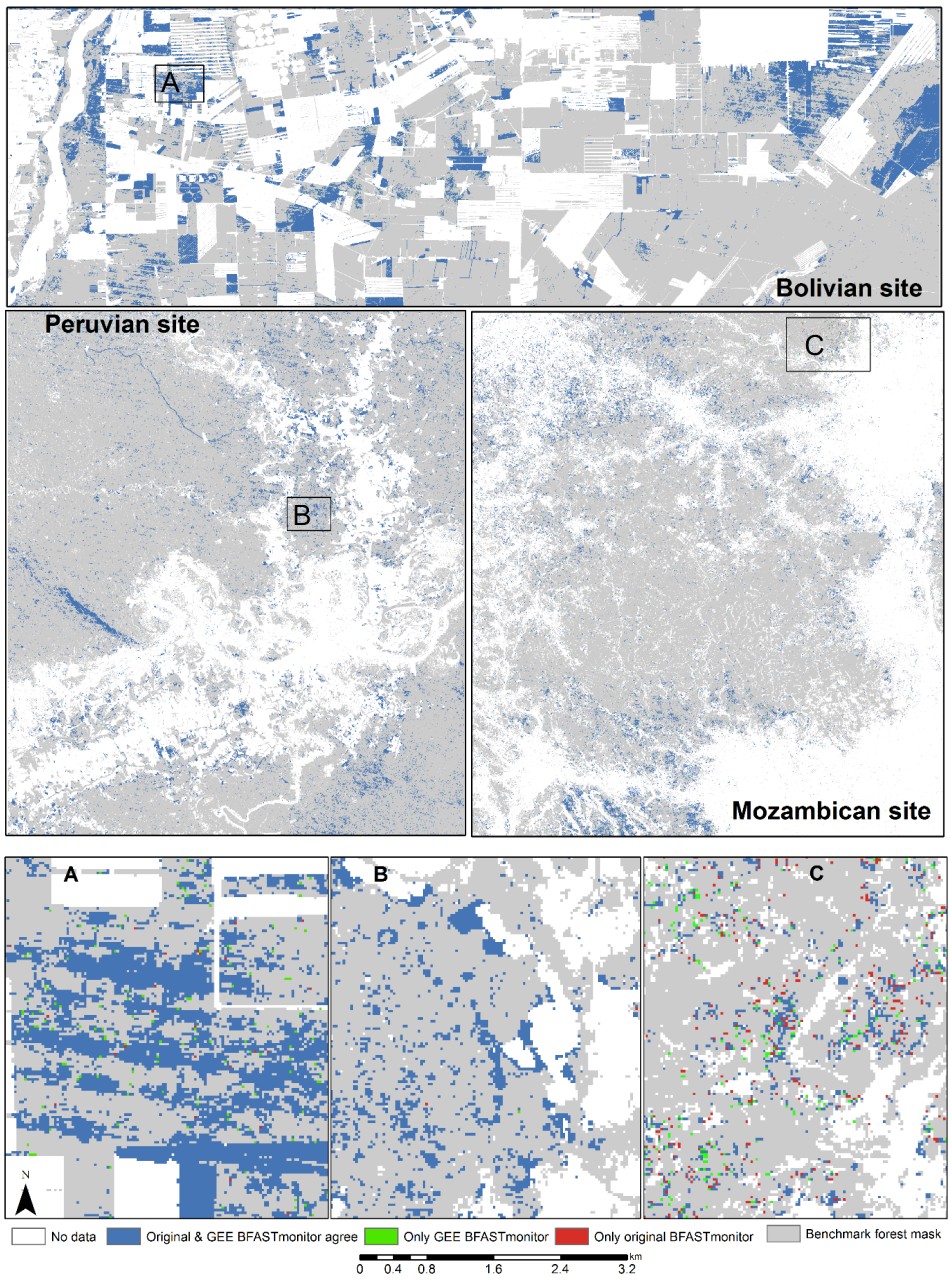

**Figure 5.** Spatial agreement and disagreement between BFASTmonitor and GEE BFASTmonitor on forest disturbance in the Landsat NDMI time series at (**A**) dry (Bolivian site) and (**B**) humid (Peruvian site) tropical forest and (**C**) Miombo woodland (Mozambican site).

There was a strong positive correlation (r = 1) in the magnitude of change between original and GEE BFASTmonitor for pixels where both implementations agreed spatially and temporally at each study site (Figure 6). This strong correlation indicates that the magnitude of change was largely similar for pixels where both implementations agree temporally on the time of forest disturbance.

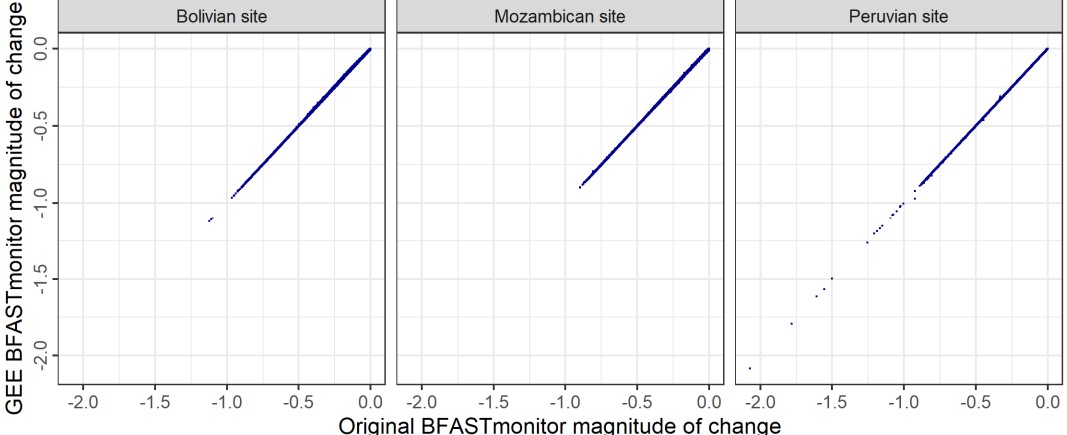

**Figure 6.** The relationship between the magnitudes of change from BFASTmonitor and GEE BFASTmonitor for forest disturbance detected at Bolivian, Mozambican, and Peruvian sites.

We observed that there were small differences in the regression model fitting between R and GEE, leading to differences in predicted values between both implementations (Figure A2, Appendix A). The differences in regression model fitting were more pronounced at the sites with strong seasonal variations (Bolivian and Mozambican site) than at the site with undefined seasonality (Peruvian site, Figure A2). The miombo woodland, in particular, had the largest differences in predicted values between original and GEE BFASTmonitor (Figure A2). Although the differences with respect to the model coefficients were relatively small (Figure A2), they led to the empirical fluctuation process of OLS-MOSUM of residuals of one the algorithm to cross the critical boundary at some pixels while the OLS-MOSUM of residuals for the other algorithm did not (Figure 8). For example, in Figure 8b, only the empirical fluctuation process of OLS-MOSUM from original BFASTmonitor that crossed the critical boundary. However, the crossing of boundary was marginal because the value of the OLS-MOSUM at the breakpoint was only 1.899832, hence slightly greater than the critical boundary of 1.897627. For GEE BFASTmonitor, however, the OLS-MOSUM value at the same point was 1.887622, and therefore was within the critical boundaries.

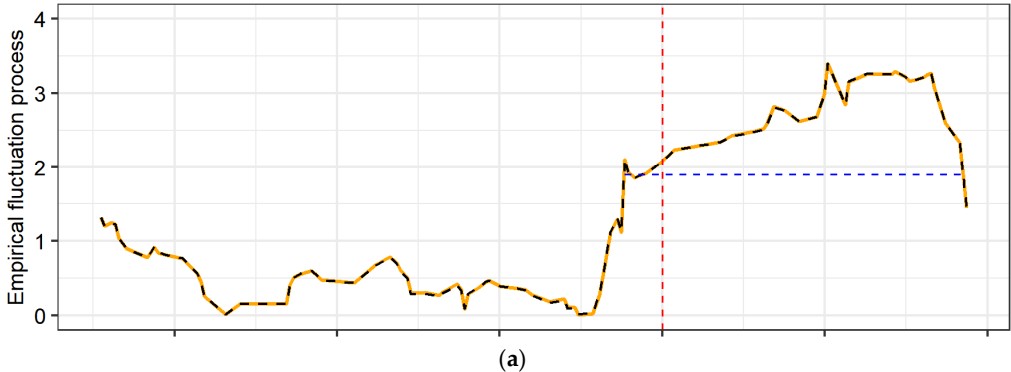

(a)

**Figure 7.** *Cont.*

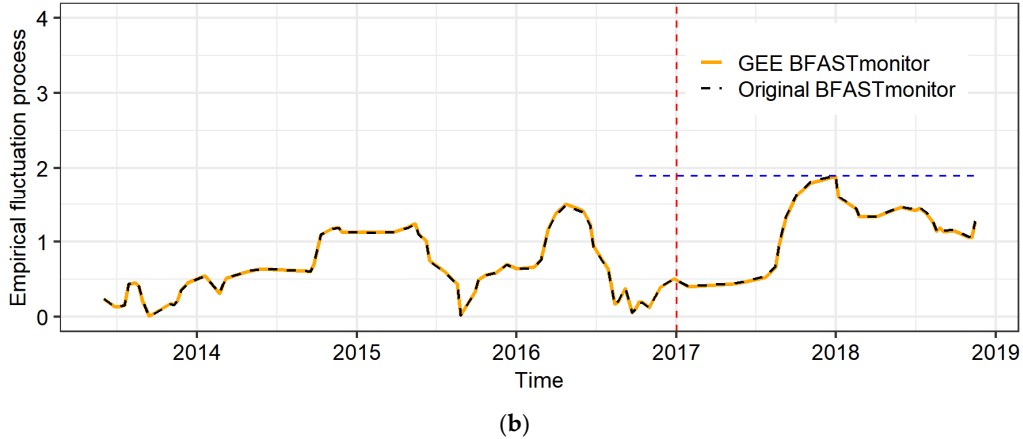

**(b)**

**Figure 8.** Absolute empirical fluctuation process for the ordinary least squares based moving sum (OLS-MOSUM) of residuals computed in R and GEE for (**a**) a pixel where both R (original BFASTmonitor) and GEE (GEE BFASTmonitor) implementations detected forest disturbance and (**b**) for a pixel where only R implementation detected forest disturbance at a Miombo woodland (Mozambican site).The vertical dashed line (red) denotes the start of the monitoring period, whereas the horizontal dashed line is the critical boundary. A breakpoint is signalled if the empirical fluctuation process crosses the critical boundary.

## 4. Discussion

Our evaluation show that we implemented BFASTmonitor on GEE correctly. This implementation offers various benefits to the remote sensing community and to those who are interested in monitoring the occurrence of abnormal changes of the earth surface using time series of earth observation data. First, BFASTmonitor has the capacity to account for seasonal variations in the time series data while detecting abnormal changes [14]. This capability allows users to distinguish abnormal changes from normal changes in the time series data at sub-annual scale. Before our implementation, the capability of sub-annual time series analysis was lacking in GEE. Unlike annual time series analysis [20], sub-annual time series analysis capability will allow users to monitor abnormal changes on earth surface in near real-time over large area. For example, those interested in forest change monitoring will be able to monitor changes also in dry tropical forests at sub-annual scale. This capability monitoring forest changes in dry tropical forest in near real-time was lacking in existing forest monitoring systems (e.g., https://www.globalforestwatch.org/). Third, by implementing BFASTmonitor on GEE, we advance the idea of bringing algorithms to data, thus eliminating costly processes of downloading and storing huge amount data. Users will not need to invest time and resources in downloading, storing and pre-processing earth observation data for them to monitor abnormal changes on earth surface using observation data.

With the original R-based implementation of BFASTmonitor, processing Landsat time series data longer than 5 years on one Landsat footprint (~32,400 km$^2$) using a standard computer with 16 GB of RAM and 8 CPU cores would take at least one week to complete. This is the reason why the R-based implementation of BFASTmonitor has not been applied to area greater one Landsat footprint to date, despite its advanced capability for identifying changes in the time series data at sub-annual scale. With our GEE implementation, however, processing the time series data longer than 4 years over one Landsat scene takes less than 5 min. Therefore, our GEE implementation will allow users to refocus their time and resources from data processing to the analysis of results to better understand how our planet is changing.

We observed small differences in the regression model fitting between R and GEE (Figure A2, Appendix A). The differences in regression model fitting explain the differences in the spatial and temporal agreement between the original and GEE BFASTmonitor. These differences in regression model fitting are not surprising, however, because such differences often occur due to the underlying solvers

approaching the computation of the regression coefficients in slightly different ways. For instance, the so-called Pseudoinverse [41] is often considered to avoid problems when the matrices of the induced linear systems of equations are singular. However, such schemes often cut off small singular values, which, in turn, might lead to different results compared to approaches that resort to normal matrix inversion (which, however, might fail when singular matrices are given). Numerical differences often occur in case the data instances are very "similar" to each other. The differences in spatial and temporal accuracy we observed between the original and GEE BFASTmonitor are minor, and we do not expect divergent conclusions between the analyses from original and GEE BFASTmonitor.

Although Landsat-7/ETM+ and Landsat-8/OLI data were used to evaluate our implementation, GEE BFASTmonitor should, with minor adjustments, be able to perform change detection on time series data from other satellite sensors (e.g., Sentinel-1 and -2). It is important for GEE BFASTmonitor to be able to analyse multi-sensor datasets because the maximum benefits for sub-annual monitoring of abnormal changes would only be properly realised when the time series is dense. In cloudy regions, in particular, temporally dense observations would only be possible when optical (e.g., Landsat and Sentinel-2) are integrated with RADAR data (e.g., Sentinel-1) into a single time series [42–45]. Therefore, a critical next step would be to enhance GEE BFASTmonitor by adding the capability for multi-source data integration. Detailed information on how apply GEE BFASTmonitor on time series data from multiple satellites will continuously be added to this webpage (https://github.com/bfast2/geeMonitor) to assist the users.

Overall, we detected many forest disturbances between 2017 and 2018 at each test site. This is expected because BFASTmonitor is a data-driven unsupervised change monitoring approach and it detects all kinds of changes in the time series [14]. Therefore, the spatial and temporal (dis)agreement between the original and GEE BFASTmonitor we report in this work does not reflect the (dis)agreement in forest cover loss detection. The users who might be interested in forest cover loss detection using GEE BFASTmonitor should perform further processing to discriminate forest cover loss from other changes (drought impact). A threshold on the magnitude of change is often used to discriminate forest cover loss from other disturbances [26], but the use of the magnitude of change enhances the omission of forest disturbances (e.g., forest degradation) with small magnitude of change [46]. To avoid this, future work should explore the potential in feeding the output of BFASTmonitor (e.g., change magnitude, the significance of change, and other temporal metrics) into other machine learning schemes, such as random forests, to differentiate forest loss from other disturbances.

In some cases, the interest is not to perform sub-annual monitoring, but to detect abrupt breaks and trends in the entire time series (e.g., [30]) to assess greening and browning patterns of vegetation [47–49] at a much higher spatial resolution. Our GEE BFASTmonitor implementation does not provide this capability, but the core component of stability testing in the time series based on the MOSUM is provided. Therefore, there is now an opportunity to build on our current implementation to also implement breakpoint and trend detection based on the BFAST approach [30].

## 5. Conclusions

We implemented BFASTmonitor algorithm, a state-of-the-art monitoring method that detects changes in time series in unsupervised manner on GEE to take advantage of the unprecedented computational power of GEE. Our implementation makes use of freely available large archives of earth observation data on GEE to support large-area and sub-annual monitoring of changes on the earth's surface, including deforestation and forest degradation. The GEE implementation agrees spatially and temporally with the original R implementation of BFASTmonitor on forest disturbance detection, implying that we have properly translated and implemented BFASTmonitor on GEE. The spatial agreement was more than 97% at each study site whereas the temporal agreement was over 94%. Small differences in regression model fitting between R and GEE inhibited perfect spatial and temporal agreement between original and GEE BFASTmonitor. With BFASTmonitor available on GEE, there is now an opportunity for efficient large-scale and sub-annual monitoring of

changes on earth's surface (e.g., deforestation) using time series data from earth observation satellites. The time series data from satellite sensors with improved spatial resolution and temporal density (e.g., Sentinel-1 and-2), which require more computational resources than coarse resolution data, can also be analysed efficiently using GEE BFASTmonitor to generate, for example, accurate and timely deforestation and forest degradation alerts. Our GEE BFASTmonitor package can, for example, be incorporated in the processing chain of existing change monitoring systems (e.g., Global forest Watch: https://www.globalforestwatch.org/ or on SEPAL (https://sepal.io/) to enhance forest change monitoring. Our GEE web application with simplified user-interface makes the GEE BFASTmonitor algorithm accessible by non-technical users who have interests in monitoring changes using satellite data.

**Author Contributions:** Implementation of algorithm, E.H. and E.E.; web application development, A.M. And E.H. and J.V.; formal analysis, investigation, and data curation, E.H. and S.R.; writing—original draft preparation, E.H.; writing—review and editing, J.V., M.H., F.G., E.E. and S.R.; funding acquisition, J.V. and M.H. All authors have read and agreed to the published version of the manuscript.

**Funding:** The initial implementation of BFASTmonitor on GEE was generously funded by Google Inc., grant number 5120906.

**Acknowledgments:** We are thankful to Noel Gorelick, and David Thau for technical assistance during the initial implementation of BFASTmonitor on GEE.

**Conflicts of Interest:** The authors declare no conflict of interest.

## Appendix A

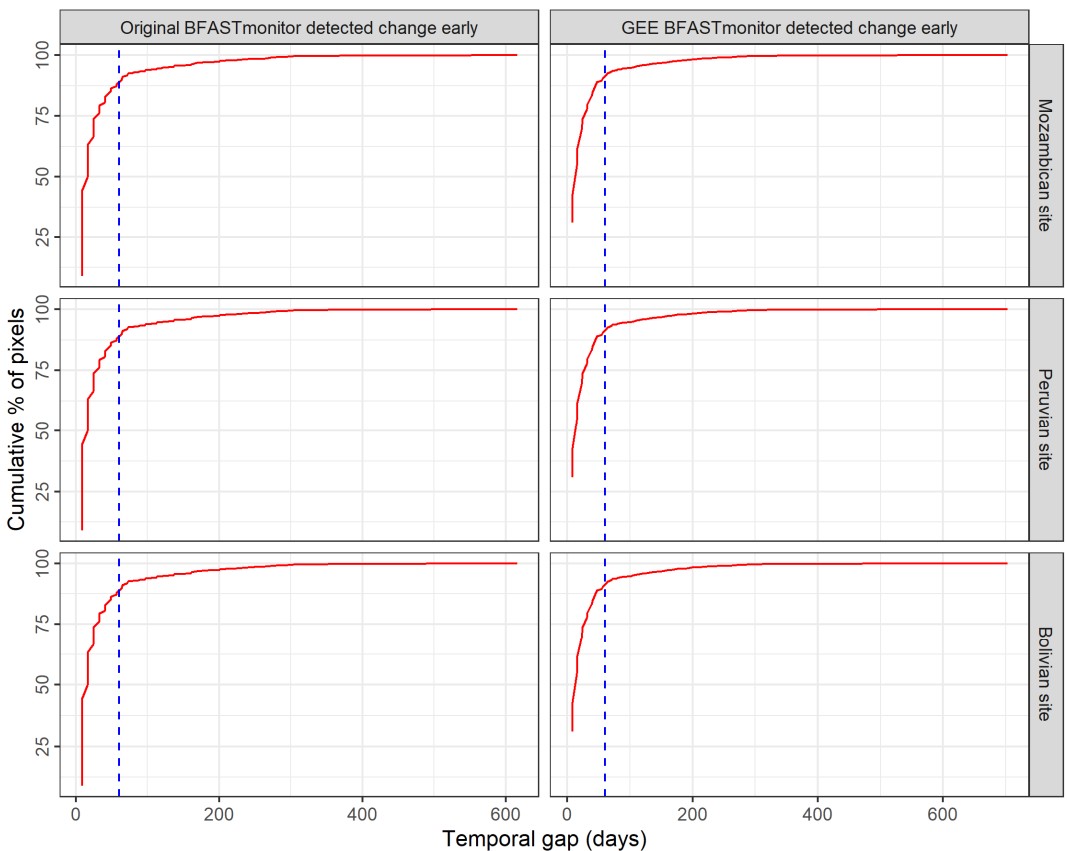

**Figure A1.** More than 85% of pixels with temporal disagreement between original and GEE BFASTmonitor at each site had disturbances detected with a temporal gap of not more than 60 days. Cumulative % of the pixels plotted against the temporal gap for pixels with temporal disagreement between original and GEE BFASTmonitor. The vertical dashed line (in blue) represents the temporal gap of 60 days.

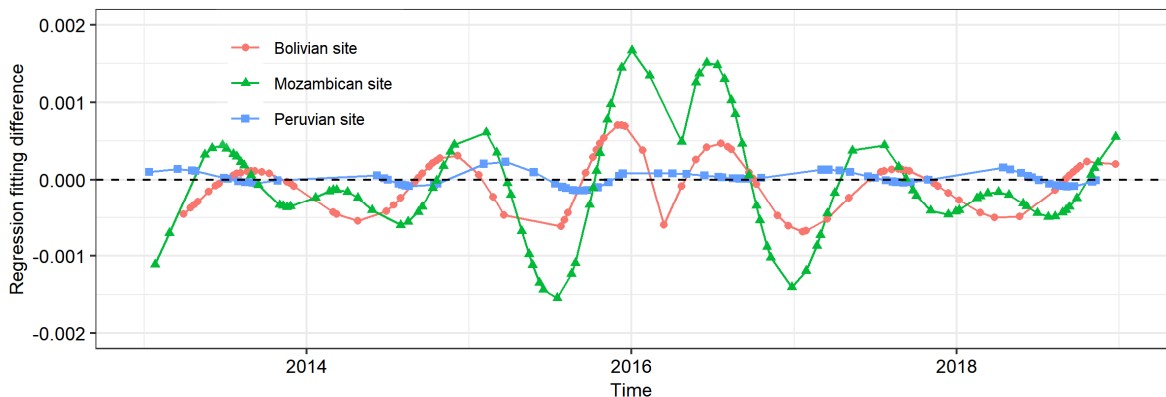

**Figure A2.** An example of regression model fitting difference between R (original BFASTmonitor) and in GEE (GEE BFASTmonitor) for the pixel time series of Landsat normalised difference moisture index at humid (Peruvian site) and dry (Bolivian site) tropical forest and Miombo woodland (Mozambican site).

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
