# Peer review of "Implementation of BFASTmonitor Algorithm on Google Earth Engine to Support Large-Area and Sub-Annual Change Monitoring Using Earth Observation Data"

_remotesensing, doi:10.3390/rs12182953_

Round 1

Reviewer 1 Report

Summary and contribution

The authors describe implementing the Breaks For Additive Season and Trend (BFAST) algorithm in Google Earth Engine. They summarize the BFAST algorithm, share links to a GEE GUI tool and some javascript code, and evaluate how their BFAST implementation GEE compares to the R implementation. They find that the GEE implementation performs similarly to their standard implementation in R, with only small differences which can be attributed to minor model differences.

Landsat time series analysis has become very popular in recent years, partly due to GEE. Making the BFAST algorithm accessible through GEE will likely increase the use of BFAST greatly, similar to how the use of LandTrendr has likely increased greatly since it has been implemented in GEE. Potential users who before would have shied away from trying to implement BFAST because of the necessity of downloading and preprocessing large amounts of data should be more likely to consider using BFAST, including me. I imagine others interested in satellite time series analysis will be interested in this paper.

General comments

If the goal is to make BFAST more accessible to users, I recommend the authors create a help page like the API page of the LandTrendr implementation in GEE (https://emapr.github.io/LT-GEE/api.html) and reference it in the paper. Here are a couple reasons why adding a help page would make BFAST more accessible:

First, even though the authors do share a link to a .js file with the BFAST code in the paper, some simple instructions in a help page explaining how to use/access the .js file in GEE would make the BFAST code even more accessible. The directions for importing the LT-GEE repository and referencing the LandTrendr.js module on the LT-GEE API page make LT-GEE very accessible. I really like how, after importing the LT-GEE repository, I can access LT-GEE in my GEE scripts with one line of code (var ltgee = require('users/emaprlab/public:Modules/LandTrendr.js'). I recommend the authors provide similar instructions/accessibility to their javascript.

Second, although the authors describe the algorithm in the paper (although this could be improved, see minor comments below), instructions describing function input parameters and outputs, like those provided in the LT-GEE API or for any R function, are currently missing. The function descriptions in the LT-GEE API page are invaluable to an outside user not familiar with the algorithm or new to GEE. Simply sharing the .js file alone, without a help page describing function inputs and outputs, is less user friendly. Adding function descriptions would make BFAST more accessible and would certainly increase its use, as it seems to have done for LandTrendr.

Some users might find pages like the authors’ page at https://andreim.users.earthengine.app/view/bfastmonitor useful, so I think it’s great the authors have created it. However, I personally didn’t find it helpful. I could be wrong, but I think many potential BFAST users would be more interested in running the BFAST algorithm function(s) in GEE themselves, than using an interface like that shown at the page above.

The paper is an appropriate length, the background has useful information and highlights the aspect of BFAST that appeals to me and likely others: subannual time series analysis. The amount of text dedicated to describing the algorithm is appropriate, although some missing details would be helpful to those not too familiar with the algorithm (see minor comments below). The authors’ evaluation is a good idea and convinces me as a user that the GEE implementation works well.

English grammar is fairly good, with only minor oddities throughout.

Minor comments

Line 24: I recommend the authors give the full name of BFAST before using the acronym.

Line 103: Should be “In this paper, we report...”

Figure 1: The figure legend is overlapped by the time series. I recommend altering the figure so they do not overlap. Also, the y-axis label “Data” is not descriptive. I recommend being more specific about what data the time series consists of.

Line 180: Should the reference be to Fig. 1 or Fig. 3a?

Lines 197-200: Did using these different approaches of calibrating to surface reflectance for Landsat-7 and Landsat-8 account for spectral differences between Landsat-7 and Landsat-8? For instance, the LandTrendr implementation in GEE uses Roy et al. 2016 to transform Landsat-8 to be consistent with Landsat-7.

Line 205: I think this is the first time that the ‘h-parameter’ is mentioned in the text. I recommend the authors describe what the h-parameter is. A help web page describing parameters would be nice for users.

Line 215: Similar to the previous comment, it’d be nice to know a reasonable range for the ‘h-parameter’. Using only one harmonic term is intuitive, or at least I think I know why using only one harmonic term makes sense, and an alpha level of 0.05 is also standard and comprehensible, but I don’t understand the ‘h-parameter’.

Line 257: How does ‘the period’ differ from the ‘start and end dates for the history and monitoring period’? A help web page describing input parameters would answer this question for users.

Figure 7. I don’t see any red line for the GEE BFASTmonitor time series.

Author Response

Please find our response to your comments in the attached document.

Reviewer 2 Report

Manuscript title: Implementation of BFASTmonitor algorithm on Google Earth Engine to support large-area and sub-annual change monitoring using earth observation data

Manuscript Number: Remote Sensing-919098

Overall: The manuscript is mostly well written, and the authors/researchers has done an excellent job in breaking the process into sections with a logical order in the manuscript. There are some questions/concerns that need to be considered during the revision.  However, the later part of the introduction needs improvement to provide a strong justification for the study. The technique/methodology seem to be sound (have some questions to be answered) and the development of a user-friendly web application is excellent. The discussion section needs improvement, especially with using references to justify the findings and providing supporting bases.

Introduction

  • At the end of the first paragraph, add a sentence on how predicted climate change in the future would make the conditions more complex and reiterate the importance of developing this tool.
  • Line 101-102: Throughout the manuscript the authors mention “small areas” and “large area”. It would be nice if it’s possible to give an indication of the extent (acres) in the introduction, when developing the justification.
  • To strengthen the justification for this study, state why large area application is important, for now and in the future. What would be the practical application? Who would be benefitting with this?
  • Figure 1: cited in the text??
  • Line 103: Re-write the first sentence, something is missing.

Materials and Methods

  • Table 1 caption: Re-word as “The size, forest type, and the number of Landsat images available for each study site”
  • Table: try not to break words in the table.
  • Question: I understand that in this study the comparison is being done between the products of GEE BFSTmonitor implementation and R-based BFASTmonitor. The classification is unsupervised. The reader might question how accurate either of these classifications are? Is it worth comparing to a supervised classification method, for a small area and assess classification accuracy?
  • Please provide a good justification for not doing such comparison
  • Line 180: citing the figure – shouldn’t it be 3a.

Results and Discussion

I would encourage the authors to improve this section by adding some references and providing a discussion. This section shouldn’t be just a presentation and an explanation of your results. There are no citations/references in the results and discussion section. Try to spread out your references. The authors neither justify the findings nor compare with previous findings. So I would recommend to go through this section again and provide supporting bases for the findings of the study. It would be better to compare your results with previous studies or other data sources from the region, to provide a better picture for the audience

  • Line 271: how significant were these differences? What would be the impact?

Outlook

  • This section needs some revision. (Would “implementation” be a better heading for this section.
  • Talk about some limitations of this study.
  • The authors mention (introduction) that disturbance detection would be challenging in areas with high seasonal variability. Does this mean this tool cannot be used or of little use in deciduous forests?
  • Even in tropics, are there any forest types that should be avoided, or would provide poor results?

Author Response

(The authors gave the same response as above.)
